# LLM-Guided Loop Bound Generation for Program Termination Verification

Zan Gong [1]  Biting Huang [1]  Fei He [1]

## Abstract

Program termination is a fundamental liveness property in software verification. Proving termination of a given program is a formidable challenge due to the undecidability of the problem. In this paper, we propose LIFT, a termination verification framework that leverages LLMs to generate loop bounds within a guess-and-check workflow. LIFT couples this generation with a sound formal validation procedure that both guarantees all reported terminations and refutes invalid loop bounds via violation analysis. Experiments on publicly accessible termination benchmarks show that LIFT significantly outperforms existing termination verification tools.

## 1. Introduction

Program termination is a critical liveness property in software verification, ensuring that a program does not run indefinitely. In general, proving termination is an undecidable problem (Turing, 1937), requiring the construction of sophisticated mathematical models, such as ranking functions and well-founded relations, to explicitly demonstrate that each loop in the program will terminate after a finite number of iterations. These models, known as *termination arguments*, are inherently difficult to generate and present a formidable challenge that often requires highly intricate and complex reasoning.

Over the past few decades, termination proving techniques have evolved into two primary categories: *white-box* and *black-box* approaches. *White-box approaches* (Heizmann et al., 2014; Giesl et al., 2014; Ströder et al., 2015; Giesl et al., 2017) exploit a program's internal structure, utilizing static program analysis and constraint solving to construct termination arguments. While these methods provide strong theoretical guarantees, they are often limited in

terms of scalability and applicability, particularly for large or complex programs. In contrast, *black-box approaches* (Xu et al., 2022; Fedyukovich et al., 2018; Urban et al., 2016) are primarily data-driven, relying on sampled data points from programs to infer candidate termination arguments. These methods use a *"guess-and-check"* framework, where a *learner* proposes candidate termination arguments, and a *teacher* validates their correctness. These methods excel in handling complex programs that lie beyond the reach of white-box techniques, but their efficiency depends heavily on the quality and representativeness of the sampled data.

With the advent of large language models (LLMs), their remarkable ability to understand and generate code naturally raises the question of whether they can serve as agents in the "guess-and-check" framework for generating candidate termination arguments. One attempt, PROTON 2.1 (Mukhopadhyay et al., 2025), leverages an LLM to synthesize ranking functions and relies on bounded model checking for validation. In contrast, LOOPY (Kamath et al., 2024) employs an LLM to generate both candidate ranking functions and their corresponding supporting invariants, constructing a formal termination proof that is subsequently verified by a solver.

To our knowledge, all existing LLM-based termination techniques, including the approaches above (Mukhopadhyay et al., 2025; Kamath et al., 2024), adopt *ranking functions* as the termination argument. A ranking function maps program states to a well-founded set (e.g., natural numbers) and must strictly decrease with each loop iteration. As termination arguments, ranking functions are relatively *fragile*: even slight inaccuracies can invalidate the entire argument. This fragility is especially pronounced for LLMs, which are prone to generating semantic errors, making the synthesis of valid ranking functions particularly challenging.

To address this issue, we shift the synthesis target from ranking functions to *loop bounds* – expressions that estimate the maximum number of iterations a loop may execute. Unlike ranking functions, loop bounds do not need to be exact; conservative over-approximations are sufficient. As a result, loop bounds provide a more stable and LLM-friendly form of termination argument. Prior work has demonstrated that loop bounds are effective as termination arguments in black-box settings (Nori & Sharma, 2013; Xu et al., 2022;

[1]School of Software, BNRist, Tsinghua University, Beijing 100084 Key Laboratory for Information System Security, MoE, China. Correspondence to: Fei He <hefei@tsinghua.edu.cn>.

*Proceedings of the 43rd International Conference on Machine Learning*, Seoul, South Korea. PMLR 306, 2026. Copyright 2026 by the author(s).

Fedyukovich et al., 2018; Urban et al., 2016).

Moreover, validating a candidate termination argument often requires providing supporting loop invariants, as the validation task is typically formulated as an assertion verification problem. One natural approach is to query an LLM to generate these invariants, as done in LOOPY (Kamath et al., 2024). However, we observe that the LLM-based loop invariant generation methods (Wu et al., 2024a; Kamath et al., 2024; Wu et al., 2024b) struggle with such validation tasks: when a candidate loop bound is invalid, LLM-based validators still attempt to construct a loop invariant to prove the verification task, even though it is inherently unrealizable.

To address this, we propose bounded violation analysis, which can, under bounded loop-unrolling, check whether a failed validation is caused by an invalid loop bound, and, if so, immediately halt unnecessary loop-invariant learning. In addition, we integrate the data-driven loop invariant synthesizer ICE-DT (Garg et al., 2016) as the primary invariant learner in our framework, providing a more robust and efficient validation procedure.

Building upon these insights, we propose Loop-bound Inference Framework for Termination (LIFT), a novel termination verification framework that leverages LLMs to generate loop bounds within a guess-and-check workflow. Our approach incorporates a sound formal validation procedure that guarantees all reported terminations and refutes invalid loop bounds through violation analysis. In addition, we transform validation outcomes into diagnostic feedbacks to guide subsequent loop bound generation. We evaluate LIFT on a publicly accessible termination benchmark suite of 171 programs, successfully solving 152 instances and outperforming existing termination verification tools.

In summary, the main contributions of this paper include:

- A termination verification framework that leverages LLMs to generate loop bounds within a "guess-and-check" workflow.

- A sound validation procedure for loop bounds that supports early refutation of invalid bounds and produces concrete diagnostic witnesses to guide LLM feedback.

- An extensive experimental evaluation demonstrating the effectiveness of our approach compared to both AI-powered techniques and conventional termination verification tools.

## 2. Preliminaries and Overview

### 2.1. Termination Verification via Loop Bounds

For simplicity, we present our approach using programs with a single loop; it generalizes to programs with multiple

```
1 assume (c >= 2);
2 while (x + c >= 0) {
3   x = x - c;
4   c = c + 1;
5 }
```

*(a)* Original Program

```
1 assume (c >= 2);
2 assume (cnt >= x + c + 1);
3 while (x + c >= 0) {
4   assert (cnt > 0);
5   x = x - c;
6   c = c + 1;
7   cnt = cnt - 1;
8 }
```

*(b)* Instrumented with a counter variable $cnt$ and a loop bound $x + c + 1$.

*Figure 1.* Program instrumentation.

loops via standard techniques (Heizmann et al., 2014; Chen et al., 2018).

Formally, let $V$ be the set of program variables and $V'$ its primed copy, representing post-state variables after a transition. Let $Init(V)$, $Guard(V)$, and $Loop(V, V')$ denote the initial condition, loop guard, and loop body, respectively. The transition relation of the loop is encoded as $Tr(V, V') \triangleq Guard(V) \land Loop(V, V')$.

A *state* $s$ is a valuation over $V$. An *execution* is a (finite or infinite) sequence of states $s_0, s_1, \ldots$ such that $s_0 \models Init$ and $\forall i \geq 0, (s_i, s_{i+1}) \models Tr$. A state $s$ is *reachable* if there exists an execution ending at $s$.

A *loop bound* $b(V)$ is an algebraic expression over $V$ that represents an upper bound on the number of loop iterations. A loop bound is *invalid* if there exists an execution that executes the loop more than $b(V)$ iterations, and *valid* otherwise. Clearly, a valid loop bound implies termination of the program.

In this setting, termination verification reduces to generating a candidate loop bound and proving its validity. Given a program $P$ and a candidate loop bound $b(V)$, validation of $b(V)$ can be reduced to a safety verification problem. Specifically, we instrument $P$ by introducing a fresh loop counter variable $cnt$, initialized to $b(V)$ and decremented at each loop iteration, and assert that the counter remains positive at the beginning of each iteration. An example of this instrumentation is shown in Fig. 1.

Let $P_{inst}$ denote the instrumented program. Its initial condition and transition relation are defined as follows:

$$Init_{inst} \triangleq Init(V) \land (cnt \geq b(V)) \qquad (1)$$

$$Tr_{inst} \triangleq Tr(V, V') \land (cnt' = cnt - 1) \qquad (2)$$

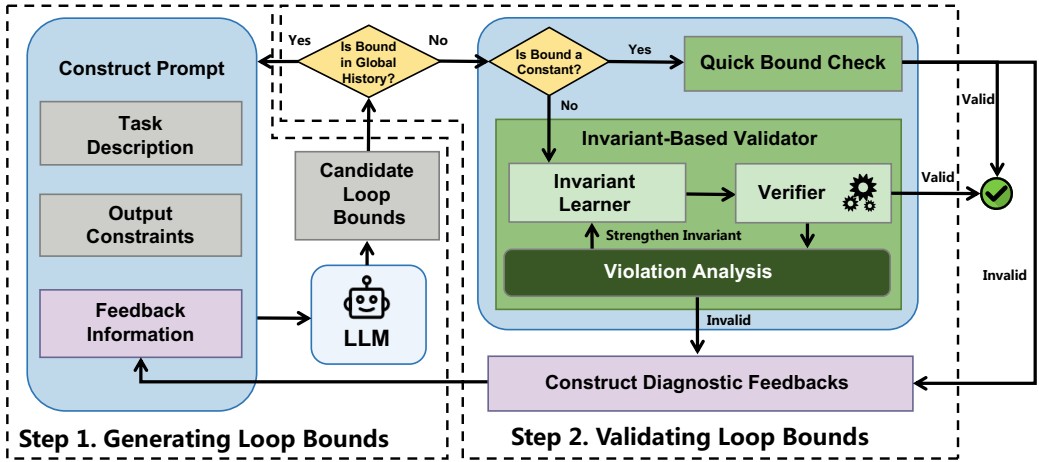

*Figure 2.* An Overview of LIFT.

The instrumented program is verified against the safety property:

$$\Psi \triangleq (cnt > 0). \tag{3}$$

If $P_{inst}$ satisfies the safety property $\Psi$, then the candidate loop bound is valid and the original program terminates.

## 2.2. Overview of LIFT

Fig. 2 presents an overview of the LIFT framework, which consists of two main components: a *loop bound generator* and a *loop bound validator*.

- **Loop Bound Generator.** This component constructs a structured prompt that includes the relevant program context, a description of the loop bound task, and output constraints, and queries the LLM to synthesize a candidate loop bound.

- **Loop Bound Validator.** This component validates the LLM-generated loop bound, either by using a quick bound checker (for constant loop bounds) or an invariant-based verifier (for symbolic loop bounds). It then analyzes the validation outcome and constructs targeted diagnostic feedback to guide the LLM in refining its next candidate.

In the following sections, we describe these two components in detail.

## 3. LLM-Guided Loop Bound Generation

This section describes how we leverage LLMs to generate candidate loop bounds.

To effectively guide the LLM, we structure the initial prompt with two core components: (1) *Task Description*, which presents the target program, defines the loop bound synthesis task, and clarifies the semantic role of loop bounds

in termination proofs; and (2) *Output Constraints*, which enforce syntactic restrictions on the generated expressions (e.g., forbidding the absolute-value operator) to ensure compatibility with the underlying validator.

Our framework supports two classes of loop bounds: (1) *Conjunctive Loop Bounds.* These use a single loop counter $cnt$ together with an upper-bound constraint expressed as a conjunction of linear arithmetic constraints over program variables (e.g., $cnt \geq x \land cnt \geq y$). This form is sufficient for most loops without complex control structures. (2) *Lexicographic Loop Bounds.* These use a tuple of counters $\langle i_0, i_1, \ldots \rangle$ that decreases according to lexicographic order, where each component is constrained by a conjunctive loop bound. This form can model more complex control flows (e.g., multi-phase loops).

We adopt a *progressive generation strategy* to synthesize loop bounds. Initially, the generator prompts the LLM to produce a simple conjunctive loop bound. The loop bound description explains that a single counter $cnt$ is decremented at each iteration, implicitly guiding the LLM to search for linear expressions or their conjunctions (conceptually, $\max(e_1, \ldots, e_n)$).

If this simple bound fails validation and cannot be corrected through the feedback iteration (Section 4.4), the generator transitions to generating a lexicographic loop bound. In this stage, the loop-bound description is significantly expanded to explicitly explain the decrement semantics of a tuple of counters (e.g., $cnt_0, cnt_1, cnt_2$). For example, the prompt specifies how the counters interact: *"The system attempts to decrement the last counter. Only when it becomes non-positive does it move to the previous counter, decrement it, and reset the subsequent ones."* This detailed semantic guidance helps the LLM better analyze complex control flows and generate correct lexicographic loop bounds.

# 4. Loop Bound Validation

Once a candidate loop bound $b(V)$ is generated, it must be validated by a sound formal procedure. Algorithm 1 outlines our core validation workflow, which distinguishes between constant and symbolic loop bounds and applies different validation strategies accordingly.

## 4.1. Validation for Constant Loop Bounds

If the generated candidate $b(V)$ is a constant, say $c$, we apply a lightweight bound-checking procedure (Xu et al., 2022) to validate it directly. Concretely, we unroll the loop body $c$ times and check the feasibility of the execution path that reaches the $(c+1)$-th iteration. If this path is infeasible, then the loop cannot execute more than $c$ iterations, and the bound is valid, implying that the program terminates. Otherwise, the bound is invalid, and the validator returns `Invalid` together with a counterexample, which is used to guide the LLM to refine the loop bound.

## 4.2. Validation for Symbolic Loop Bounds

For symbolic bounds, the procedure invokes an invariant-based validator. The core of this validator is the synthesis of a *safe loop invariant* – a predicate that must hold at the loop head for every iteration and is strong enough to establish the desired safety property.

Formally, given a safety verification task $\langle P_{inst}, \Psi \rangle$ (as defined in Section 2.1), a safe loop invariant $Inv(V_{inst})$ must satisfy the following three conditions:

$$Init_{inst} \implies Inv(V_{inst}) \quad (4)$$
$$Inv(V_{inst}) \wedge Tr_{inst} \implies Inv(V'_{inst}) \quad (5)$$
$$Inv(V_{inst}) \wedge Guard(V_{inst}) \implies \Psi \quad (6)$$

These conditions are referred to as *Reachability* (Eq. (4)), *Inductiveness* (Eq. (5)), and *Safety* (Eq. (6)), respectively.

The invariant-based validator consists of two components: an *Invariant Learner* (LEARNER), which proposes candidate invariants, and a *Verifier* (VERIFIER), which checks whether they satisfy the three conditions above. Given a candidate invariant $I$:

- If $I$ violates Reachability (Eq. (4)) or Inductiveness (Eq. (5)), the verifier returns `Invalid_Inv` together with a counterexample $ce$. This indicates that $I$ is not a valid invariant, and $ce$ is fed back to LEARNER for refinement.

- If $I$ satisfies Reachability and Inductiveness but violates Safety (Eq. (6)), the verifier returns `Failure` with a counterexample state $ce$. This case requires further analysis, which we detail in Section 4.3.

---

**Algorithm 1:** Loop Bound Validation Procedure

**Input:** Program $P$, Candidate Loop Bound $b(V)$
**Output:** `Valid`, `Invalid`($s_{ce}$), or `Timeout`

1 **if** *is_constant(b)* **then**
2      **return** QUICKBOUNDCHECK($P, b$)

3 $P_{inst} \leftarrow$ INSTRUMENT($P, b$)
4 $\Lambda \leftarrow \emptyset$
5 **while** $\neg$ TIMELIMITEXCEEDED() **do**
6      $I \leftarrow$ LEARNER($P_{inst}, \Lambda$)
7      $res, ce \leftarrow$ VERIFIER($I, P_{inst}, \Psi$)
8      **if** $res = Success$ **then**
9          **return** `Valid`
10      **else if** $res = Invalid\_inv$ **then**
11          $\Lambda \leftarrow \Lambda \cup \{ce\}$
12      **else if** $res = Failure$ **then**
13          $s_{ce} \leftarrow$ VIOLATIONANALYSIS($P_{inst}, ce$)
14          **if** $s_{ce}$ **then**
15              **return** `Invalid`($s_{ce}$)
16          **else**
17              $\Lambda \leftarrow \Lambda \cup \{ce\}$

18 **return** `Timeout`

---

- If $I$ satisfies all three conditions, the verifier returns `Success`. In this case, $I$ is a safe loop invariant that proves the safety property $\Psi$, and the loop bound $b(V)$ is thus validated.

To further improve the effectiveness of validation, we integrate the *k-induction* technique following Chen et al. (Chen & He, 2021), which applies the $k$-induction principle to termination proofs. This technique facilitates invariant inference by strengthening the base case while relaxing the inductiveness requirements on the invariant.

## 4.3. Violation Analysis

When the verifier returns `Failure`, we perform *violation analysis* to determine whether the failure is caused by an *invalid loop bound* or by a *weak invariant*.

More precisely, a loop bound is invalid if it underestimates the number of loop iterations, in which case there exists at least one reachable state that violates the safety property $\Psi$ (defined in Eq. (3)). In this situation, synthesizing a safe loop invariant that establishes $\Psi$ is fundamentally impossible, and the correct action is to promptly refute the invalid loop bound rather than futilely attempting to refine the invariant.

However, most LLM-based loop invariant generation frameworks (Wu et al., 2024a; Kamath et al., 2024) lack the ability to recognize such *logically unrealizable* verification tasks. When the loop bound is invalid, they repeatedly attempt to strengthen the invariant, unaware that the specification itself is inconsistent. This leads to substantial computational

waste and provides no meaningful diagnostic feedback.

To address this issue, we perform *bounded violation analysis* whenever the verifier returns `Failure` together with a counterexample state $s_{ce}$. Concretely, we unroll the loop from 0 to $m$ steps and, at each depth $k$, check the satisfiability of the following formula:

$$Init_{inst}(s_0) \wedge \bigwedge_{j=0}^{k-1} Tr_{inst}(s_j, s_{j+1}) \wedge (s_k = s_{ce})$$

If this formula is satisfiable for some $k \leq m$, then $s_{ce}$ is a *violation witness*, which confirms the existence of a real violation and proves that the loop bound is invalid. In this case, invariant learning is aborted and the validator returns $Invalid(s_{ce})$. If the formula is unsatisfiable for all $k \leq m$, we conservatively treat $s_{ce}$ as potentially spurious and return it to the invariant learner for further refinement. Since this procedure is bounded, it may fail to refute invalid loop bounds whose violations only manifest after more than $m$ iterations.

To further improve robustness, we also integrate a conventional data-driven invariant learner, ICE-DT (Garg et al., 2016; Xu et al., 2020), into our framework. ICE-DT maintains three kinds of samples: positive samples (reachable states), negative samples (error states), and implication samples (state transitions). Instead of relying on a single bounded trace, ICE-DT confirms violations by constructing a valid transition path from a positive sample to a negative sample using implication samples. If such a path exists, it constitutes a definitive proof that a violation exists and yields a concrete violation witness.

We implement both LLM-based invariant synthesis and ICE-DT as invariant-learning backends in our framework; by default, we use ICE-DT as the invariant learner due to its strong capability in producing concrete violation witnesses.

### 4.4. Diagnostic Feedbacks

The validation outcomes are fed back to the LLM in a structured form to guide its subsequent attempts at generating improved loop bounds.

**Diagnostic Feedback for Violation.** When the validation procedure returns `Invalid` together with a concrete violation witness $s_{ce}$, we construct a feedback prompt that explicitly informs the LLM that the proposed loop bound is invalid and incorporates the witness $s_{ce}$ into the context. This concrete state provides precise information about the program's reachable state space and helps the LLM refine its subsequent hypotheses.

In addition, during our experiments, we observed that LLMs often struggle with precise constant-offset reasoning in loop bounds. To mitigate this issue, we introduce a *Const-Offset*

*Correction (COC)* heuristic before continuing the feedback loop. When validation returns `Invalid`, the system automatically attempts to relax the failed bound by adding a small constant offset $C$. If the adjusted bound passes validation, program termination is successfully proven, thereby significantly reducing unnecessary refinement iterations.

**Diagnostic Feedback for Timeout.** A `Timeout` result typically arises from two main causes. (1) *High Complexity*: the generated loop bound expression may be syntactically complex or define a semantically large upper bound; in either case, the validator incurs a heavy computational burden and may fail to complete within the time limit. (2) *Semantic mismatch with loop progress*: the generated loop bound may be syntactically simple but still uninformative if it ignores the variables that actually determine loop progress. For example, in Fig. 1, a valid bound depends on both $x$ and $c$ (e.g., $x+c+1$), whereas the LLM may propose a bound that depends only on $c$ (e.g., $c+3$). However, the loop transition implies $x'+c' = x+1$, indicating that the relevant progress measure is coupled to $x$ (and more precise expressions such as $x + c$ rather than to $c$ alone. In such cases, even though the proposed bound is simple, the invariant learner may fail to connect the counter property (e.g., $cnt > 0$) to the loop guard, leading to many futile refinement rounds during invariant generation and potentially a timeout. In these situations, we instruct the LLM to simplify the expression and to generate a loop bound that is explicitly coupled to the key variables of the loop.

**Diagnostic Feedback for Syntax Error.** If the generated loop bound does not conform to the grammar (e.g., contains illegal operations or undeclared variables), the system catches this and feeds back the error expression along with specific syntax rules to request a format correction.

## 5. Evaluation

We evaluate LIFT to answer the following questions:

**RQ1 (Effectiveness and Efficiency):** How does LIFT compare with state-of-the-art termination verification tools in terms of both effectiveness and efficiency?

**RQ2 (Validation Strategy Analysis):** How do different validation strategies affect the performance of LIFT?

**RQ3 (LLM Interaction Strategy Analysis):** How do different LLM interaction strategies (including progressive generation and diagnostic feedback) affect the performance of LIFT?

### 5.1. Experimental Setup

*Benchmarks and Platform.* We use the benchmark suite proposed by (Fedyukovich et al., 2018), which consists of 171 terminating programs. All experiments are conducted

*Table 1.* Performance comparison of LIFT with conventional and LLM-based termination verification tools. "LIFT Only" – instances solved exclusively by LIFT (Gemini); "Baseline Only" – instances solved exclusively by the corresponding baseline.

| Tools | LIFT | | PROTON 2.1 (2025) | Loopy (2024) | | ddlTerm (2022) | UAutomizer (2014) | FreqTerm[1] (2018) |
| --- | --- | --- | --- | --- | --- | --- | --- | --- |
| | Gemini | DeepSeek | Llama-1B (FT) | Gemini | DeepSeek | | | |
| #Solved | **152** | 139 | 144 | 125 | 89 | 134 | 106 | 92 |
| LIFT Only | - | 14 | 23 | 36 | 67 | 19 | 55 | 65 |
| Baseline Only | - | 1 | 15 | 9 | 4 | 1 | 9 | 5 |
| Avg. T. on Solved (s) | 189.4 | **114.9** | 147.5 | 140.8 | 183.0 | 13.9 | 8.0 | 4.4 |

on a machine equipped with an AMD EPYC 7H12 (2×64-core) CPU and 1.0 TiB RAM.

*Baselines.* We compare LIFT against both LLM-based and conventional termination verifiers. For LLM-based baselines, we choose LOOPY (Kamath et al., 2024) and PROTON 2.1 (Mukhopadhyay et al., 2025), two recently proposed LLM-based termination verifiers. For conventional termination verifiers, we select UAUTOMIZER (Heizmann et al., 2014), FREQTERM (Fedyukovich et al., 2018), and DDLTERM (Xu et al., 2022), all of which represent state-of-the-art tools in termination analysis.

*Configuration.* All baseline tools are run with their default settings, with FREQTERM configured to use Spacer as its underlying verifier. For LIFT, we enable the progressive generation strategy with at most 5 iterations for conjunctive loop bound generation and 10 iterations for lexicographic loop bound generation. Symbolic loop bounds are validated by an invariant-based backend, which uses ICE-DT as the default invariant learner and Boogie backed by SMT solving as the verifier, with a 60-second timeout for the validator.

*Metrics.* Effectiveness is measured by the number of solved benchmarks, and efficiency is measured by the average runtime over solved instances (denoted as "Avg. T. on Solved"). In addition, for a more fine-grained comparison, we report the number of instances solved exclusively by each tool, as well as a breakdown of the runtime spent on loop bound generation and validation.

### 5.2. RQ1: Effectiveness and Efficiency

We first evaluate the overall performance of LIFT compared with the baselines. Table 1 summarizes the results. In this part, LIFT uses ICE-DT as the default validator (as mentioned at the end of Section 4.3).

For a fair LLM-based comparison, we run LIFT and LOOPY under the same two off-the-shelf LLMs (Gemini-2.5-Flash and DeepSeek-V3.1), using each model's default configuration (Gemini with thinking enabled by default, and DeepSeek with thinking disabled by default). For PROTON 2.1, since it relies on a fine-tuned Llama model released by its authors, we directly use their fine-tuned model (shown as Llama-1B (FT) in the table).

We use LIFT (Gemini) as our default configuration and compare it against each of the other tools, including LIFT (DeepSeek) and all baseline verifiers. The row "LIFT Only" reports the number of benchmarks solved by LIFT (Gemini) but not by the corresponding column configuration, while the row "Baseline Only" reports the number of benchmarks solved by the corresponding column configuration but not by LIFT (Gemini).

**Overall effectiveness.** LIFT (Gemini) solves 152 benchmarks, outperforming all conventional and LLM-based tools. Compared to the best conventional tool (DDLTERM), LIFT achieves a significantly higher success rate, demonstrating the advantage of leveraging LLMs' code understanding capabilities for loop bound hypothesis generation.

**Comparison with LLM-based baselines.** Compared to PROTON 2.1, LIFT solves 8 more instances. Although PROTON 2.1 can handle a relatively large number of examples with high efficiency, its validation mechanism (bounded check with $k = 3$) provides weaker soundness guarantees. In contrast, LIFT couples LLM-based loop bound generation with a formal verifier that discharges the underlying safety verification obligations in an SMT-based manner, ensuring that all reported terminations are formally sound.

The exclusive-solved statistics also indicate that LIFT (Gemini) solves many instances that each baseline fails to solve, while the converse is much rarer, suggesting that LIFT provides strictly stronger coverage rather than trading off a different subset of instances.

**Efficiency.** In terms of the average runtime on solved instances, LLM-based approaches are generally less efficient than conventional tools, largely due to the inherent latency of LLM inference. Moreover, LIFT can be somewhat slower than LOOPY when using Gemini-2.5-Flash, since LOOPY follows a non-iterative workflow: it generates multiple candidate ranking functions in one round and validates them separately, while LIFT performs multiple refinement iterations under validation-driven feedback. Nevertheless, the overall runtime of LIFT remains within an

---

[1] FREQTERM was run with a compatibility patch: since its expected older Z3 version was unavailable, we disabled an incompatible interface that would otherwise cause execution to abort.

*Table 2.* Comparison with iterative-feedback LOOPY.

| Method | Solved | Avg. T. on Solved (s) |
|---|---|---|
| LOOPY | 125 | 140.8 |
| LOOPY+Iter | 139 | 336.2 |
| LIFT | 152 | 189.4 |

*Table 3.* Comparison of LIFT with different invariant-based validators.

| Backend Config | Solved | Inf. T (s) | Val. T (s) |
|---|---|---|---|
| LIFT + ICE-DT | **152** | 32265 | 32499 |
| LIFT + LaM4Inv | 120 | 35235 | 33517 |
| LIFT + LEMUR | 100 | 93709 | 52059 |

acceptable range for practical use, and the iterative design does not incur prohibitive overhead: on DeepSeek-V3.1, LIFT achieves better efficiency than the other two LLM-based termination verification tools. Overall, these results indicate that LIFT provides a favorable balance between effectiveness and efficiency.

**Comparison with LOOPY+simple feedback iteration.** Although LOOPY is non-iterative, it generates multiple candidate ranking functions in a single round and validates them separately. Therefore, LOOPY can also be viewed as using multiple trials, but these trials are not guided by validation feedback. To make the comparison more controlled and further rule out the possibility that LIFT's advantage comes merely from more trials, we equip LOOPY with a simple feedback iteration loop. We refer to this experimental configuration as LOOPY+Iter. This configuration converts the result returned by LOOPY's own validator into a natural-language explanation and sends it back to the LLM for ranking-function revision. The feedback indicates which verification conditions are not proved, such as non-negativity, monotonic decrease, or lexicographic decrease. We use the same model, Gemini-2.5-Flash, and the same two-stage iteration budget as LIFT: 5 iterations in the first stage and 10 iterations in the second stage.

As shown in Table 2, adding a similar feedback loop improves LOOPY, increasing the number of solved benchmarks from 125 to 139. This confirms that iterative feedback is indeed useful. However, LOOPY+Iter still remains clearly weaker than LIFT, which solves 152 benchmarks. Moreover, iterative feedback makes LOOPY substantially slower, increasing its average time on solved benchmarks to 336.2 seconds, compared with 189.4 seconds for LIFT. These results suggest that LIFT's gains are not merely due to more trials; rather, they indicate that the termination verification workflow combining our loop-bound generation with the validation procedure is more effective.

**Analysis of Unsolved Cases.** We further inspect the 19 benchmarks not solved by the default LIFT configuration. The failures fall into two broad categories. In 10 cases, the LLM does not generate a sufficiently precise loop bound within the iteration budget; the proposed bounds are repeatedly refuted by counterexamples, especially for programs with nondeterministic phase changes, resets, or piecewise control flow. In the remaining 9 cases, validation is blocked by the current validator capability: they induce proof obligations that the invariant learner cannot discharge within the budget. These results suggest that the remaining limitations come from both candidate-bound precision and the validator's support for complex bound expressions.

### 5.3. RQ2: Validation Strategies Analysis

Our default validation strategy uses ICE-DT as the invariant learner. To evaluate the impact of different validation strategies, we replace ICE-DT with two LLM-based invariant learners, and report the resulting overall performance.

**LIFT + LaM4Inv:** We adopt the invariant learning procedure of LaM4Inv (Wu et al., 2024a) as invariant learner for invariant synthesis [2]. Since LaM4Inv does not provide violation analysis by default – a crucial component for refuting invalid loop bounds – we augment it with our Bounded Violation Analysis (BVA) strategy (Section 4.3). In this configuration, we set $m = 30$ and use a 20-second timeout for each bounded check.

**LIFT + LEMUR**: We use the LEMUR framework (Wu et al., 2024b) (powered by ESBMC (Gadelha et al., 2019)) for invariant synthesis. It provides bounded violation analysis by leveraging ESBMC's Bounded Model Checking (BMC) feature.

Both LaM4Inv and LEMUR methods were originally designed and evaluated with GPT-family models, so we kept the original model family. Specifically, we use GPT-5, a more recent and advanced GPT-family model, for invariant generation in LaM4Inv and LEMUR. To ensure a fair comparison, Gemini-2.5-Flash is retained for loop bound inference across all configurations.

Results are reported in Table 3, where columns "Inf. T (s)" and "Val. T (s)" denote the total time spent on loop bound inference and validation, respectively, aggregated over all instances (both solved and unsolved).

The results show that the ICE-DT backend achieves the best overall performance, outperforming other configurations in both the number of solved benchmarks and total verification time. This superiority is primarily driven by

---

[2] Since the original LaM4Inv implementation targets its own SMT-format benchmarks, we reimplemented it in a LIFT-compatible manner.

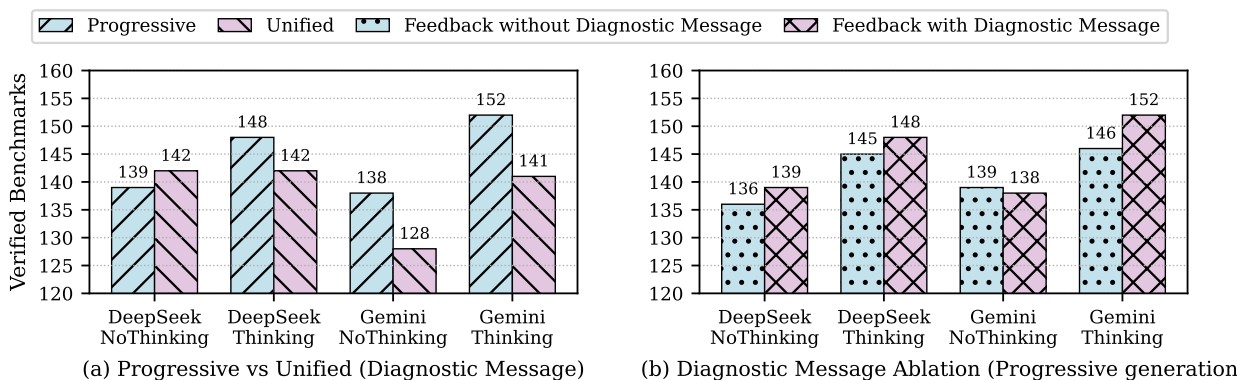

*Figure 3.* Ablation results for interaction strategies. (a) Progressive generation (Conjunctive → Lexicographic) vs. unified generation (with diagnostic feedback enabled). (b) Diagnostic feedback ablation under progressive generation.

ICE-DT's powerful *violation analysis* capability. In our experiments, we observed that both LaM4Inv (equipped with BVA) and LEMUR failed to refute certain invalid loop bounds that were successfully refuted by ICE-DT. Furthermore, we found that even with BVA enabled, LaM4Inv still struggles to synthesize invariants strong enough to validate correct loop bounds, and LEMUR exhibits similar limitations. This suggests that, while violation analysis is important, the ability to generate sufficiently strong invariants remains equally indispensable.

### 5.4. RQ3: LLM Interaction Strategy Analysis

This experiment evaluates the impact of our LLM interaction strategies, including the progressive generation strategy and the diagnostic feedback mechanism. We conduct the study using two LLMs: Gemini-2.5-Flash and DeepSeek-V3.1.

**Progressive Generation Strategy.** We compare our *progressive generation strategy* (Conjunctive → Lexicographic) against a *Unified Generation* strategy where the LLM autonomously selects the bound form and generate concrete loop bounds. For a fair comparison, the Unified strategy is allocated a maximum of 15 iterations (matching the total budget of the progressive generation approach).

As shown in Fig. 3(a), the progressive generation strategy generally brings some improvement. Without it, the model often attempts complex bounds prematurely, leading to timeouts. Overall, these results suggest that this progressive structure is a useful default strategy, although its impact varies across configurations.

**Efficacy of Diagnostic Feedback.** We perform an ablation study by removing the diagnostic feedback components (Violation Witness and timeout diagnosis) while keeping the remaining feedback prompt structure unchanged. As shown in Fig. 3(b), diagnostic feedback is generally helpful in the our framework. The validation-derived messages

provide useful guidance for refining failed candidates.

## 6. Related Work

### 6.1. Termination Analysis

Proving termination typically involves synthesizing a Ranking Function that strictly decreases during execution. Traditional white-box tools, such as APROVE (Giesl et al., 2017), and ULTIMATE AUTOMIZER (Heizmann et al., 2014), utilize static analysis techniques like term rewriting and trace abstraction. While rigorous, these methods often struggle with scalability and complex syntax.

Other approaches adopt "guess-and-check" strategies. FRE-QTERM (Fedyukovich et al., 2018) employs Syntax-Guided Synthesis (SyGuS) to generate loop bound candidates based on syntactic templates. In contrast, DDLTERM (Xu et al., 2022) is a data-driven approach that relies on data traces provided by the underlying verifier; it applies convex optimization to synthesize loop bounds in affine forms. LIFT aligns with the "guess-and-check" structure but replaces syntactic templates or geometric optimization with LLMs to hypothesize complex loop bounds.

### 6.2. LLMs for Formal Verification

The application of LLMs to formal verification has expanded rapidly, covering various aspects from specification generation to invariant synthesis.

**Program Specification and Domain-Specific Verification.** Several works focus on generating formal specifications. AUTOSPEC (Wen et al., 2024) utilizes static analysis to decompose programs into sub-tasks, synthesizing distinct specification forms for each. SPECGEN (Ma et al., 2025) employs a mutation-based method to assist LLMs in generating program specifications. In domain-specific contexts, SMARTINV (Wang et al., 2024) introduces a "Tier

of Thought" strategy to fine-tune and prompt the LLM for verifying smart contracts, while LAUREL (Mugnier et al., 2025) uses Retrieval-Augmented Generation (RAG) to generate assertions for Dafny programs.

**Invariant Generation and Termination.** The primary use of LLMs in verification remains the synthesis of loop invariants. Representative methods such as CLAUSE2INV (Cao et al., 2025) and LAM4INV (Wu et al., 2024a) leverage an LLM to propose candidate predicates, and then apply specific filtering and combination strategies to select and compose them into an inductive invariant. Additionally, (Pei et al., 2023) explores fine-tuning techniques to evaluate the intrinsic capability of LLMs in invariant generation tasks. LOOPY (Kamath et al., 2024) is the work most closely related to ours, as it extends LLM-based verification to termination analysis. LOOPY prompts the LLM to guess a ranking function and its supporting invariants to prove termination. However, it operates as a one-time attempt for termination: if the guessed ranking function cannot be validated, it is simply discarded without any feedback loop to refine the guess.

A critical challenge facing these methods is the "refutation gap" – when the property to be proved (e.g., a loop bound or a specification) is invalid, most LLM-based invariant learners often iterate fruitlessly rather than reporting a violation. While LEMUR (Wu et al., 2024b), which is an LLM-based invariant learner as well, could partially address this by leveraging its backend verifier, ESBMC (Gadelha et al., 2019), which can refute false assertions. However, LEMUR is constrained by the bounded unrolling depth, making it less effective for termination proofs where violations may occur deep in execution. LIFT overcomes these limitations by integrating LLMs with a powerful validation procedure which is able to refute invalid loop bounds effectively.

## 7. Discussion

**Limitations.** Although LIFT demonstrates strong performance, future work could further improve its efficiency and robustness. The current framework relies on LLM inference and multi-round interaction between loop-bound generation and formal validation; while this design improves verification coverage, reducing the cost of these interactions through more efficient candidate prioritization or feedback scheduling would make the framework more practical for larger-scale verification tasks. In addition, some candidate bounds may introduce nonlinear arithmetic obligations that are challenging for current SMT-based verifiers and invariant learners. Strengthening the arithmetic reasoning backend or integrating specialized nonlinear reasoning techniques is a promising direction for improving robustness on complex programs.

**Conclusion.** We propose LIFT, a novel framework that utilizes the LLM to generate multi-type loop bounds in a progressive manner for program termination verification. LIFT constructs a feedback loop between the LLM and an effective validation procedure equipped with the ability of violation analysis. This combination allows LIFT to effectively refute invalid loop bounds and, crucially, to translate these validation outcomes into diagnostic feedback, enabling the LLM to iteratively refine its loop bound hypotheses. We conduct extensive experiments to validate the efficacy of our approach. Our experiments show that LIFT significantly outperforms the state-of-the-art termination verification methods.

## Impact Statement

This paper presents work whose goal is to advance the field of Formal Verification. There are many potential societal consequences of our work, none which we feel must be specifically highlighted here.

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

# A. Prompt Templates

This appendix presents the concrete prompt templates used in our framework. It includes: (1) the conjunctive loop bound query prompt; (2) the lexicographic loop bound query prompt; and (3) the validation-driven feedback prompts tailored to different validation outcomes (Invalid/Violation with witness, Timeout, and Syntax Error).

---

**System Instruction**

You are an expert in formal verification and program analysis, specializing in termination proofs.

Your task is to analyze the provided program code and infer a suitable loop bound to help prove its termination.

---

**User Prompt (Core Content)**

**Context:** We are trying to prove the termination of a `while` loop in a program. The program uses an implicit ranking-function approach, where a counter `i` is decremented in each iteration. The loop bound will replace the placeholder `assume(%M:i%);`.

**Loop Bound Description:** The loop bound `m(X)` is an expression over program variables `X` (e.g., `x, y`) and must be an *overapproximation* (never smaller than actual iterations). It is used as: `assume(i >= m(X));`. Example outputs include:
`assume(i >= x + 10);`   `assume((i >= 1) && (i >= y));`

**Task:** Based on the provided code, generate a suitable loop bound to replace `assume(%M:i%);` and output only the bound condition(s).

**Output Constraints (Notes):**
(1) Each bound is an `assume` inequality with `i` on the left-hand side and `>=`.
(2) `i` must NOT appear on the right-hand side of the inequality.
(3) Conjunction `&&` is allowed; disjunction `||` is forbidden.
(4) Multiple `assume` statements are allowed and must end with semicolons, e.g., `assume(i >= x+10); assume(i >= y);`.
(5) No division (`/`, `div`) and no nonlinear ops (`|x|`, `abs`).

**Program Code:** `[program_code]`

**Output:** Just provide the loop bound condition. Don't explain.

---

*Figure 4.* Conjunctive loop bound query prompt.

**System Instruction**

You are an expert in termination proofs for programs, skilled at identifying *lexicographical* loop bounds.

Your task is to analyze the code and generate a suitable *lexicographical loop bound* in a specific format.

**User Prompt (Core Content)**

**Context:** Simple/conjunctive bounds were insufficient; we now consider *Lexicographical Loop Bounds*.

**Lexicographical Loop Bounds Explained (Semantic Scaffold):** Termination relies on a tuple of expressions. The system models this using multiple implicit counters `i0, i1, ..., ik` (supported up to `i2`). The system attempts to decrement the last counter. Only when it becomes non-positive does it move to the previous counter, decrement it, and reset the subsequent ones.The prompt includes an explicit sketch of decrement/reset logic, e.g.

```
if (i2 > 0) { i2 := i2 - 1; }
else if (i1 > 0) { i1 := i1 - 1; havoc i2; assume(i2 >=
m2(X)); }
else { i0 := i0 - 1; havoc i1, i2; assume(i1 >= m1(X));
assume(i2 >= m2(X)); }
```

**General Properties:** Bounds are overapproximations; `m(X)` should be linear/affine; avoid `/`, `div`, `|x|`, `abs`.

**Task:** Infer needed counters (`i0, i1, ...`) and their bounds in terms of program variables.

**Program Code:** `[program_code]`

**Output Format (Strict):** Output exactly one line:
```
assume(i0 >= m0(X) && i1 >= m1(X) && ...);
```
No other text. Loop counters must not appear on the right-hand side of the inequality.

*Figure 5.* Lexicographic loop bound query prompt.

**Base Prompt (unchanged)**

Either Page 1 (conjunctive query) or Page 2 (lexicographic query) is used as the base. The system then appends a *diagnostic feedback block* depending on the validator outcome.

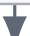

**Type A: Syntax / Grammar Error**

**Type Explain:** `Your previous answer ... contains a grammatical error.`

**Hints (excerpt):** Ensure `i >= m(X)` with `i` on the left-hand side. Avoid division and nonlinear ops. Do not use undeclared vars. Loop counter must not appear on the right-hand side of the inequality.

**Type B: Verification Failure (Bound Too Small)**

**Type Explain:** `... the loop bound you provided is too small ...`

**Optional Evidence:** `The following is a counterexample given by the verifier: [ce].`

**Hints (excerpt):** The bound must be an *overapproximation* and never smaller than actual iterations.

**Type C: Timeout**

**Type Explain:** `... likely a timeout.`

**Hints (excerpt):** Timeout may come from (1) overly complex bounds, or (2) bounds semantically disconnected from loop variables. Suggest simplifying and coupling to key variables so invariants are easier to infer.

**Common Constraint Added After Feedback**

`***The answer you provide next must be different with your previous answer!***`
`**Just give your answer. Don't explain.**`

**Assembly Rule in the Script**

The final prompt is formed by concatenation:
`prompt_content := base_prompt_content + feedback_block`

*Figure 6.* Validation-driven feedback prompt add-on.

## B. Token Consumption

Table 4 reports token consumption for representative LIFT configurations. Each configuration name follows the pattern LIFT_*Model*_*Mode* optionally followed by *nodiagn*. Here, *Model* is either DeepSeek-V3.1 or Gemini-2.5-Flash; *thinking/nothinking* indicates whether the LLM reasoning mode is enabled; and *nodiagn* denotes the ablation setting where validation-derived diagnostic feedback is removed. "Avg. Total Tok. (All Files)" denotes the average total token consumption per benchmark file over the entire benchmark set, while "Avg. Total Tok. (Solved Files)" is computed only over successfully solved files.

*Table 4.* Token consumption under different LIFT configurations.

| Config | Solved Files | Avg. Total Tok. (All Files) | Avg. Total Tok. (Solved Files) |
|---|---|---|---|
| LIFT_DeepSeek_nothinking_nodiagn | 136 | 3,767 | 1,685 |
| LIFT_DeepSeek_nothinking | 139 | 4,378 | 2,231 |
| LIFT_DeepSeek_thinking_nodiagn | 145 | 41,079 | 22,020 |
| LIFT_DeepSeek_thinking | 148 | 46,608 | 27,852 |
| LIFT_Gemini_nothinking_nodiagn | 139 | 3,980 | 1,983 |
| LIFT_Gemini_nothinking | 138 | 4,954 | 2,425 |
| LIFT_Gemini_thinking_nodiagn | 146 | 33,548 | 16,199 |
| LIFT_Gemini_thinking | 152 | 38,319 | 21,423 |

