# OpenReview forum: "LLM-Guided Loop Bound Generation for Program Termination Verification"
_ICML.cc/2026/Conference — ICML 2026 regular_

### Official Review · Reviewer_wDa3 · 2026-03-09

**Soundness:** 4
**Presentation:** 4
**Significance:** 3
**Originality:** 3
**Overall Recommendation:** 4
**Confidence:** 4

**Summary:**

This paper proposes a method and a tool for verifying termination of programs based on LLMs. First step consists of prompting an LLM to generate t bounds-expression that upper bounds the number of executions of a loop. Checking of the correctness of this bound can be phrased as a problem of checking loop invariants. For this second step, either an LLM can be used, or classical symbolic methods can be used. A key challenge is to handle failures of this second step as a feedback to revise the propsoed bounds expressions. Experimental evaluation consists of comparing this new method to prior methods (including calling LLMs directly and symbolic methods for termination analysis) on an existing benchmark set both in terms of ability to prove termination and computational effort needed.

**Compliance With Llm Reviewing Policy:**

Affirmed.

**Final Justification:**

The discussion has been useful. I continue to find the direction promising, I will keep my score

**Key Questions For Authors:**

1. There is a detailed discussion of prompting strategy of using LLMs to propose loop bounds. When it comes to using LLMs to propose invariants to check validity of loop bounds, there is not much discussion. Authors claim that ICE+DT works the best, but did you try different prompting strategies for LLM-based approach to work for this phase. Alternatively, can ICE+DT be extended to generate loop bounds? Or is there some intrinsic reason that LLMs work well only for one of these two tasks.
2. Maybe I missed it, but the benchmark suite contain programs that do not terminate. Is your method able to establish non-termination?

**Limitations:**

Yes

**Strengths And Weaknesses:**

Strength:
+ Proving termination of programs is a long-studied, challenging, and important problem in software analysis, and the paper makes tangible progress on this.
+ Uses loop bounds, rather than more traditional ranking functions. This seems to work better.
+ Experimental results, while limited to a single benchmark suite, do show significant improvements.

Weakness:
 While the combination of techniques is interesting, individual components are well studied.

---

> ### Author Rebuttal · Authors · 2026-03-31
>
> Thank you for the careful reading and constructive feedback. We clarify questions on invariant generation, and non-termination below.
>
> > There is a detailed discussion of prompting strategy of using LLMs to propose loop bounds. When it comes to using LLMs to propose invariants to check validity of loop bounds, there is not much discussion. Authors claim that ICE+DT works the best, but did you try different prompting strategies for LLM-based approach to work for this phase. Alternatively, can ICE+DT be extended to generate loop bounds? Or is there some intrinsic reason that LLMs work well only for one of these two tasks.
>
> Loop-bound generation and invariant generation are fundamentally different tasks for LLMs: loop bounds are integer expressions, whereas invariants are logical Boolean formulas. Many prior works, already cited in our paper, study LLM-based invariant synthesis. In this paper, our focus and main contribution are on loop-bound generation, so in the validation stage we intentionally reused relatively mature LLM-based invariant-generation methods from prior work rather than designing a new prompt specifically for invariant synthesis.
>
> For the same reason, ICE-DT cannot be directly extended to loop-bound generation. It is designed to learn logical invariants, not integer expressions, and this mismatch in output type already prevents a direct extension.
>
> We do not mean that LLMs are useful for only one of these two tasks, since both are important in our framework. Rather, the relative comparison to traditional methods appears different: InvBench [1] and our Fig. 3 both suggest that current LLM-based invariant synthesis still trails strong traditional tools. In contrast, LLM-guided loop-bound generation can be more effective than traditional methods because it can reason over the full loop semantics to propose a global upper-bound expression, rather than being limited by incomplete sampled states. The intrinsic reason behind this difference is an interesting question that we plan to investigate further.
>
> > Maybe I missed it, but the benchmark suite contain programs that do not terminate. Is your method able to establish non-termination?
>
> LIFT does not establish non-termination. Our framework is specifically designed for termination proving via candidate loop bounds and their formal validation. Proving non-termination is a different task and typically requires different proof objects (e.g., recurrent sets), as in prior work on non-termination analysis [2], rather than ranking/bound-style certificates. We therefore do not claim support for non-termination proving in the current paper. We will revise the paper to make this limitation clearer.
>
> Accordingly, the 171-program benchmark we evaluate on contains only terminating programs, which is also consistent with the setting in the source benchmark paper.
>
> Thank you again for the helpful feedback.
>
> [1] Wei et al. *InvBench: Can LLMs Accelerate Program Verification with Invariant Synthesis?* arXiv 2025.
> [2] Han et al. *Data-driven Recurrent Set Learning For Non-termination Analysis*. ICSE 2023.

---

> > ### Author Rebuttal · Reviewer_wDa3 · 2026-04-01
> >
> > Thank you for the rebuttal. Indeed I understand that establishing non-termination will require very different techniques, so the fact that LIFT cannot establish non-termination is not a fatal flaw. I will stay with my current score.

---

### Official Review · Reviewer_os7m · 2026-03-11

**Soundness:** 2
**Presentation:** 3
**Significance:** 2
**Originality:** 2
**Overall Recommendation:** 2
**Confidence:** 3

**Summary:**

The paper introduces LIFT, an LLM-guided framework for predicting program termination by synthesising LLM-generated loop bounds and iterative invariant-based validation diagnostic feedback. LIFT uses an LLM-based generator with a validator using an invariant learner. On the 171 benchmark, LIFT outperforms other LLM-based(LOOPY, PROTON2.1) and conventional termination verifier(UAUTOMIZER, FREQTERM) baselines.

**Compliance With Llm Reviewing Policy:**

Affirmed.

**Key Questions For Authors:**

(1) have you tried real-world code with complex loops, control flow, pointers, heap and interprocedural calls?

(2) How did you validate the loop bounds? Are you using SMT-solvers or custom-built tools?

(3) In which specific scenarios does the LLM guidance provide a statistically significant boost that justifies its computational cost over the pure ICE-DT approach?

(4) For the 10 cases where LIFT+ICE-DT failed, was the bottleneck the LLM inference or ICE-DT’s failure?

**Limitations:**

I did not found discussions on limitations of this work.

**Strengths And Weaknesses:**

Strength:

(1) LIFT shows significantly higher performance than the baselines; it solves 161 out of 171 benchmark problems than the other  LLM-based baselines (Loopy 125/171) and conventional tools (ddlterm 134/171)

(2) The integration of “violation analysis” is a practical solution that prevents the system from getting stuck in case of futile invariant learning loops.


Weakness:

(1) Upon inspecting the benchmark samples, I found that most of them contain simple loop logics. Not sure how the framework can be extended in real-world settings.

(2) As the invariant validator component, the authors described details about the invariant learner (ICE-DT). But I am not sure how the authors validated the synthesized loop bounds.

(3) Though the paper claims to be “LLM-guided” bound generation, the framework's success is heavily dependent on ICE-DT (as Table 2 shows).

(4) Even on simpler loops, LIFT is significantly slower than the baselines.

---

> ### Author Rebuttal · Authors · 2026-03-31
>
> Thanks for the careful reading and constructive feedback. We clarify questions on scope and validation below.
> > have you tried real-world code with complex loops, control flow, pointers, heap and interprocedural calls?
>
> We have not tried these complex programs, and our benchmark does not include such real-world programs. We tried our best to collect usable benchmarks from the literature, and this is the most widely recognized public benchmark we could identify for this setting; its source traces back to the SV-COMP termination category, one of the most authoritative competitions in termination verification. More broadly, termination proving is classically undecidable and intrinsically hard, and most prior work still focuses on programs of this style [1-2]. Our goal here is to establish, on this community-accepted benchmark, that LLM-guided loop-bound generation improves termination proving over both prior LLM-based and conventional baselines.
> Looking forward, we believe the framework can be extended to richer settings such as multiple or nested loops using techniques in [3]. This could be our future work.
> > How did you validate the loop bounds? Are you using SMT-solvers or custom-built tools?
>
> We clarify the loop-bound validation procedure below.
> Given a candidate loop bound $b(\mathbf{x})$, we first instrument the original program to obtain $P_{\text{inst}}$ (see Fig. 1(b)), introducing an auxiliary counter variable. In $P_{\text{inst}}$, we add an assertion that ensures this counter never drops below zero; the validator then checks whether this assertion always holds.
> At this point, validating the loop bound reduces to proving a safety property of $P_{\text{inst}}$. Specifically, an invariant learner (typically ICE-DT) is used to synthesize a loop invariant for $P_{\text{inst}}$ that satisfies the three conditions in Sec. 4.2 (Eqs. (4), (5), and (6)). These conditions are then checked by a verifier—in our implementation, the Boogie verifier backed by SMT solving, which is also the native verifier used together with ICE-DT. If the verifier confirms all conditions, we have obtained a valid invariant proving that the assertion in $P_{\text{inst}}$ always holds. This in turn ensures that the instrumented counter variable $cnt$ never goes below zero, so the candidate bound is indeed a valid loop bound. Therefore, loop bound validation is precisely the process of finding a valid invariant for the instrumented program $P_{\text{inst}}$.
> > The framework's success is heavily dependent on ICE-DT. In which specific scenarios does the LLM guidance provide a statistically significant boost that justifies its computational cost over the pure ICE-DT approach?
>
> We acknowledge that ICE-DT is effective for invariant generation. However, the focus of our work is loop bound generation, where we show that the LLM-guided approach achieves better performance than conventional methods. The reliance on ICE-DT for invariants is a separate component, and improving LLM-based invariant generation remains future work.
> A key point here is that ICE-DT itself is an invariant learner, not a termination prover. By itself, it does not take source code and directly produce a termination argument. The contribution of LIFT is precisely to use the LLM to propose semantically meaningful loop bounds from the code, and then use formal validation to prove or refute them.
> > For the 10 cases where LIFT+ICE-DT failed, was the bottleneck the LLM inference or ICE-DT’s failure?
>
> Based on our inspection, the remaining 10 failures are not attributable to a single bottleneck. In 4 cases, the LLM does not generate a sufficiently aligned loop bound within the allowed iterations; in other 6 cases, validation remains difficult, e.g., due to nonlinear arithmetic obligations or because ICE-DT cannot synthesize a strong enough invariant within the budget. Overall, the dominant failure mode is repeated validation timeout in the guess-and-check loop. We will add this discussion to the limitations section.
> > Even on simpler loops, LIFT is significantly slower than the baselines.
>
> The time overhead of LIFT mainly comes from LLM inference and the multi-round iterative process. Nevertheless, LIFT remains more efficient than prior LLM-based baselines while retaining strong capability, and it solves more programs than even the strongest traditional data-driven baseline by leveraging global semantic understanding beyond incomplete sample data.
>
> Thank you again for the constructive feedback.
>
> [1] Fedyukovich et al. *Syntax-guided termination analysis*. CAV 2018.
> [2] Xu et al. *Data-driven loop bound learning for termination analysis*. ICSE 2022.
> [3] Garg et al. *ICE: A robust framework for learning invariants*. CAV 2014.

---

> > ### Author Rebuttal · Reviewer_os7m · 2026-04-02
> >
> > Your rebuttal answered my questions well. Thank you. However, I do think this technique needs much help from external tools, and the types of programs and loops addressed are simple. This paper does not have much contributions in machine learning. So I will keep my score.

---

> > > ### Author Response · Authors · 2026-04-07
> > >
> > > We thank the reviewer for the thoughtful follow-up comments. Below we respond briefly to the remaining concerns point by point, while clarifying the intended scope and contribution of the paper.
> > >
> > > - this technique needs much help from external tools：
> > >     - Our paper is not intended as a purely end-to-end neural verification method. Rather, it targets  the application of ML/LLMs to a difficult formal verification problem. In this setting, the use of external formal tools is not a weakness - it is a necessary component for preserving soundness. The contribution of our paper lies in showing how LLMs can be used effectively within this soundness-critical setting: specifically, to generate loop bounds in an LLM-friendly form and to refine them through validation-driven diagnostic feedback.
> > >
> > > - the types of programs and loops addressed are simple;
> > >     - We have made every effort to collect the most widely recognized benchmarks from public sources, spanning from SV-COMP to those used in the most related literature [1-2]. Furthermore, the proposed technique can be extended to handle more complex programs and loops, as demonstrated in [3]. Therefore, the types of programs and loops explored in the benchmark set should not be a big issue.
> > >
> > > - this paper does not have much contributions in machine learning.
> > >     - The paper was submitted to the primary area of Applications, which is precisely intended for work that applies ML methods to solve important problems. From that perspective, using LLMs to make substantial progress on an important task in program verification - namely, termination verification - is itself a meaningful ML contribution.
> > >
> > > We thank the reviewer again for the thoughtful follow-up comments.

---

### Official Review · Reviewer_iUtB · 2026-03-11

**Soundness:** 3
**Presentation:** 3
**Significance:** 2
**Originality:** 2
**Overall Recommendation:** 4
**Confidence:** 4

**Summary:**

This paper presents LIFT, an LLM-based termination verifier. Unlike previous LLM-based approaches in this area, which typically synthesize ranking functions, the proposed method focuses on discovering and verifying loop bounds. The LLM is asked to propose candidate loop bounds, and each proposal goes through an iterative validation and feedback loop. Instantiated with Gemini as the LLM and ICE-DT as the invariant learner, LIFT is evaluated on 171 termination programs. The results show that LIFT outperforms two existing LLM-based termination verifiers as well as several traditional approaches.

**Compliance With Llm Reviewing Policy:**

Affirmed.

**Final Justification:**

The authors’ responses address my concerns regarding experimental details and ablation studies. Although I still feel the framework is relatively straightforward and that the experiments are mostly conducted on small programs, I will keep my positive score.

**Key Questions For Authors:**

- LIFT is an iterative system, with 5 iterations for conjuective loop bound generation following by 10 lexigraphic ones. The authors mention LOOP is a single-pass system. How about PROTON2.1? Is it possible the better performance of LIFT mainly comes from more trials? The authors claim that the loop bounds are more "LLM-friendly" than ranking functions. Have the authors tried generating ranking functions with a similar iterative feedback loop? It would also be helpful to see performance as a function of the number of trials.

- In 4.2 the authors listed possibilites of different validation outcomes, but verifier timeouts or unknown is not mentioned. There is some discussion of timeouts in Section 4.4, but the overall handling of timeouts remains unclear. For example, in Algorithm 1, it appears that only the entire validation procedure has a timeout. Does the verifier itself have a timeout? What about the Violation Analysis step? In the experiments, only the timeout for ICE-DT is mentioned. What about other components?

- Other than LLM-based and ICE-DT invariant learner, have the authors tried traditional methods such as Spacer for generating invariants?

- How is the bound m set in the experiment?

**Limitations:**

yes

**Strengths And Weaknesses:**

## Strength

- The overall system design is reasonable and well justified. Within in the feedback loop, the proposed loop bounds can be conjunctive or lexicographic and are validated using different procedures. The system also attempts to exploit information returned by symbolic tools, such as distinguishing whether a failed validation is caused by an invalid loop bound or by a weak invariant.
- The paper is well written. The motivation is clear, and both preliminary concepts and related work are presented clearly.
- The proposed system outperforms exisiting systems on one benchmark set.

## Weakness

- Other than the LLM component for proposing loop bounds, the framework itself appears fairly standard for termination verification.
- Fedyukovich et al. (2018) benchmarks seem to be mostly simple programs. Even in their paper, they evaluated on additional larger-scale benchmarks. It would be interesting to see whether the proposed LLM-based approach can help with more practical and larger programs.
- The description about the "by ... Only" metrics are confusing.
- I also have some questions/concerns about the system design and result presentation listed in the Questions section.

---

> ### Author Rebuttal · Authors · 2026-03-31
>
> Thank you for the careful reading and constructive feedback. We clarify questions on LIFT's gains, timeout settings, and evaluation scope below.
>
> > Is LIFT's advantage mainly due to more trials rather than the loop-bound formulation itself?
>
> We believe that the gain of LIFT is not mainly due to "more trials." Both Loopy and PROTON2.1 are essentially single-pass frameworks in the sense that they generate a batch of candidate ranking functions once and then validate them respectively, without using validation feedback to iteratively revise the candidates. In contrast, LIFT generates one candidate at a time, validates it, and uses the resulting diagnostic information to guide the next proposal. So the key difference is not simply the number of attempts, but the presence of a feedback-driven repair loop.
>
> To directly address your question, we have also conducted a preliminary comparison by adding a similar iterative feedback loop to Loopy under the Gemini setting. The resulting comparison is summarized below:
>
> | Method | Solved Benchmarks | Avg. Time on Solved (s) |
> | --- | ---: | ---: |
> | Loopy + Gemini | 125 | 140.8 |
> | Loopy + Gemini + similar iterative feedback | 139 | 336.2 |
> | LIFT + Gemini | 161 | 220.2 |
>
> These results support two points. First, adding a similar feedback loop does improve Loopy's capability, increasing the number of solved benchmarks from 125 to 139. However, it still remains clearly weaker than LIFT + Gemini, which solves 161 benchmarks. This further supports our central claim that loop-bound generation is more LLM-friendly than general ranking-function generation, while iterative feedback itself is indeed useful and can materially improve performance. We will report this comparison more clearly and in more detail in the revision.
>
> > Timeout handling and validation outcomes.
>
> In the current implementation, when ICE-DT is used as the learner inside the validator, the timeout reported in the paper is effectively the overall validator timeout. The verifier is not given a separate independent global timeout beyond this learner-side budget, because ICE-DT relies on incremental interaction between learner and verifier: if the verifier does not return a result for the current round, the learner cannot update the sample set and continue.
>
> Regarding violation analysis, the situation depends on the learner:
>
> * With ICE-DT as the learner, the violation-analysis step is lightweight in practice and terminates quickly.
> * With LLM-based invariant learners, we use Bounded Violation Analysis, where the bound is set to (m=30) and each bounded check has a timeout of 20 seconds.
>
> We will revise the paper to distinguish these cases more explicitly and make the timeout settings much clearer.
>
> > Other than LLM-based and ICE-DT invariant learner, have the authors tried traditional methods such as Spacer for generating invariants?
>
> We have not tried Spacer, as it is designed for Constraint Horn Clause (CHC)-based verification, which differs from our current encoding setting. The front-end and verification condition encoding between CHC-based and SMT-based verifiers are not directly interchangeable.
>
> > How is the bound m set in the experiment?
>
> In our experiments, we set (m=30) for Bounded Violation Analysis. This value is chosen empirically: it is usually sufficient to expose shallow violations while keeping the bounded checks efficient. We agree that this parameter should be stated explicitly in the implementation details.
>
> > Fedyukovich et al. (2018) benchmarks seem to be mostly simple programs. Even in their paper, they evaluated on additional larger-scale benchmarks. It would be interesting to see whether the proposed LLM-based approach can help with more practical and larger programs.
>
> We checked the additional larger-scale benchmarks referenced in [1], but the public link is currently inaccessible, so the benchmark suite used in our paper is the most representative public dataset we could reliably obtain and reproduce. Please also see our response to the 1st question of Reviewer os7m, where we discuss this scope limitation in more detail.
>
> > The description about the "by ... Only" metrics are confusing.
>
> Thank you for pointing out that this metric description was unclear. Here:
>
> * **by LIFT(Gemini) Only** means the number of instances solved by LIFT(Gemini) but not by the tool in the corresponding column;
> * **by Other Only** means the number of instances solved by the tool in the corresponding column but not by LIFT(Gemini).
>
> We will rewrite the table caption and the surrounding text to make this set-difference interpretation explicit.
>
> Thank you again for the constructive feedback.
>
> [1] Fedyukovich et al. *Syntax-guided termination analysis*. CAV 2018.
> [2] Xu et al. *Data-driven loop bound learning for termination analysis*. ICSE 2022.

---

> > ### Author Rebuttal · Reviewer_iUtB · 2026-03-31
> >
> > Thank you for the response, and I am keeping the score.

---

### Official Review · Reviewer_GgPm · 2026-03-12

**Soundness:** 3
**Presentation:** 2
**Significance:** 3
**Originality:** 3
**Overall Recommendation:** 4
**Confidence:** 4

**Summary:**

The paper presents an LLM-aided verifier LIFT to check the termination of a given program by estimating its loop bound. The loop bound it returns, if any, is guaranteed to be valid, hence establishing a sound termination proof for the given program. Experiments with 171 terminating programs manifest the advantage of LIFT in terms of effectiveness, with affordable efficiency, in comparison with state-of-the-art LLM-aided and classical termination verifiers.

**Compliance With Llm Reviewing Policy:**

Affirmed.

**Final Justification:**

My concerns have been addressed as expected, hence my score remains the same.

**Key Questions For Authors:**

1. What does the name LIFT stand for?
2. If the generated loop bound is encapsulated into a ranking function, would a ranking-function based verifier benefit from this loop bound, with or without the diagnostic feedbacks?
3. Missing details about the implementation and experiments, e.g., how does k-induction fit LIFT? What's the setting of $m$ for bounded violation analysis? How does the performance of LIFT vary with different configurations? Apart from the run-time consumptions, how about the token consumptions?
4. How efficient is LIFT in handling unsolved and non-terminating programs, in comparison with others?

**Limitations:**

Typical threats to the validity of this work needs to be discussed, esp. analyzing why LIFT fails to handle the remaining 10 terminating programs.

**Strengths And Weaknesses:**

Strength
+ An alternative perspective to prove program termination, leveraging the code reasoning capabilities of LLMs for direct loop bound generation
+ Well-designed experiments, demonstrating the significant performance of LIFT and its core components

Weaknesses
- The established loop bound can be regarded as a special form of a ranking function on the additional counter variable(s). Thus, this perspective actually still follows the typical one, but with a more direct representation.
- Diagnostic feedbacks are a typical way to work with LLMs, while orchestration details are missing, e.g., in aligning an input program with its stateful semantics for and by LLMs.
- Line 128: $P_{inst} \vDash \Psi$ needs a more clear and careful definition.
- It is also not clear what is the conceptual role of the max function in loop bound generation?
- The usage and settings of LLMs seem cherry-picking, although default settings of LLMs are used, Gemini is used in its thinking mode, but DeepSeek is not, while the recent GPT-5 is used only for RQ2. Thus, a systematic use of these LLMs with fair mode settings is preferred.
- Only terminating programs are evaluated. 176 non-terminating programs, from the same source, shall also be included to evaluate to what extent invalid loop bounds can be refuted by LIFT and others.

---

> ### Author Rebuttal · Authors · 2026-03-31
>
> Thank you for the careful reading and constructive feedback. Below we clarify the main conceptual and implementation issues you raised.
> > What does LIFT stand for?
>
> LIFT stands for Loop-bound Inference Framework for Termination.
> > If the generated loop bound is encapsulated into a ranking function, would a ranking-function based verifier benefit from this loop bound, with or without the diagnostic feedbacks?
>
> Yes. A ranking-function-based verifier can benefit directly from the generated loop bound. By instrumenting the program with an additional counter variable - initialized to the bound and decremented appropriately - the bound itself becomes a valid ranking function. This holds regardless of diagnostic feedback, as the bound alone encodes the termination argument.
>
> > Missing details about the implementation and experiments, e.g., how does k-induction fit LIFT? What's the setting of 'm' for bounded violation analysis? How does the performance of LIFT vary with different configurations? Apart from the run-time consumptions, how about the token consumptions?
>
> In our experiments, k-induction is only used in the invariant-based loop-bound validation stage: rather than synthesizing a standard 1-inductive invariant, we synthesize an invariant for a k-inductive termination certificate, which strengthens the base case and relaxes preservation, reducing the burden on invariant synthesis [1]. Besides, our bouded violation analysis uses `m=30`. For token consumption, representative configurations show that no-thinking uses about 4k average total tokens per file, while thinking uses about 30k-50k. We will include a concise appendix table about it in our revision.
> > How efficient is LIFT in handling unsolved and non-terminating programs, in comparison with others? To what extent invalid loop bounds can be refuted by LIFT and others?
>
> Briefly, we did not conduct a separate comparative evaluation on non-terminating programs, because LIFT - like other termination verification tools (e.g., [2]) - is designed for termination proving. For non-terminating programs, no valid loop bound exists, so LIFT simply times out and reports "unknown". Including such cases would primarily measure timeout behavior rather than the effectiveness of bound inference, which is the focus of our evaluation.
> Besides, the ability to refute invalid loop bounds can also be demonstrated with terminating programs, because before the final loop bound is validated, each iteration already includes refuting invalid candidate bounds. In RQ2, we also show that LaM4Inv and LEMUR fail to refute some invalid loop bounds that are successfully refuted by ICE-DT.
> > Diagnostic feedbacks are a typical way to work with LLMs, while orchestration details are missing, e.g., in aligning an input program with its stateful semantics for and by LLMs.
>
> The detailed prompt and feedback information for the LLM interaction is provided on the last page of the appendix. We will point readers to this location more explicitly in the revision.
> > The usage and settings of LLMs seem cherry-picking...
>
> Although the configurations presented in the main experiments are limited, Figure 3 already reports the thinking/no-thinking comparison for both gemini and deepseek. We present only deepseek without thinking in the main results because the latency overhead is substantial. Since our evaluation setting requires multi-round interaction, this overhead accumulates quickly, and it is especially burdensome for Loopy, which relies on the LLM both for ranking-function generation and for invariant generation. For fairness in the main comparison, we therefore use dpsk in its default no-thinking configuration.
> GPT-5 appears only in RQ2, for the invariant-generation baselines LaM4Inv and LEMUR. The main reason is faithful reproduction: both methods were originally designed and evaluated with GPT-family models, so we kept the original model family. We will clarify this in the revision.
> > Line 128: $(P_{\text{inst}} \models \Psi)$ needs a more clear and careful definition..
>
> Our definition around $(P_{\text{inst}} \models \Psi)$ is not sufficiently rigorous, and we will define it explicitly in the revision.
>
> > It is also not clear what is the conceptual role of the max function in loop bound generation?
>
> We present '`max` operator to explain that when loop-bound expressions are generated conjunctively, the conjunction should be understood semantically as a piecewise `max` of those expressions.
> > Remaining 10 unsolved terminating cases and limitations.
>
> Due to the word limit, please see our response to the 4th question of Reviewer os7m, where we discuss in more detail.
>
> Thank you again for the insightful feedback. We believe these clarifications will significantly improve the paper.
>
> [1] Chen et al. *Proving termination by k-induction*. ASE 2020.
> [2] Xu et al. *Data-driven loop bound learning for termination analysis*. ICSE 2022.

---

> > ### Author Rebuttal · Reviewer_GgPm · 2026-04-02
> >
> > Thanks for the response, and I will keep the score accordingly.

---

### Decision · Program_Chairs · 2026-04-30

**Decision:**

Accept (regular)

**Comment:**

Reviewers agreed that this paper addresses an interesting research problem in LLM-guided program verification: how to use LLMs-guided techniques to generate and validate loop bounds. The proposed framework for the verification of loop bounds is well designed and integrates well LLM-guided loop bound generation, with more traditional invariant generation. The evaluation shows good improvement with respect to other approaches. The reviewers and the area chair agree that the results of this paper are good contribution for the ICML community.